# Can neoadjuvant chemoradiotherapy be omitted in cT2N+ and cT3 mid-rectal cancer: Protocol for a prospective, observational, cohort study (CANO)

Feza Karakayali[1], Cigdem Arslan[2‡*], Tayfun Bisgin[3‡], Ilknur Erenler Bayraktar[4], Onur Bayraktar[4], Aras Emre Canda[5]

1 Department of Surgery, Faculty of Medicine, Baskent University, Istanbul, Türkiye, 2 Department of Surgery, Istanbul Health and Technology University, Istanbul, Türkiye, 3 Department of Surgery, Faculty of Medicine, Dokuz Eylul University, Izmir, Türkiye, 4 Department of Surgery, Faculty of Medicine, Halic University, Istanbul, Türkiye, 5 Department of Surgery, Acibadem Kent Hospital, Izmir, Türkiye

☯ These authors contributed equally to this work.
‡ CA and TB also contributed equally to this work.
* cigdemarslan@hotmail.it

## Abstract

### Background

Neoadjuvant chemoradiotherapy (nCRT) followed by total mesorectal excision (TME) is the standard treatment for locally advanced rectal cancer. However, nCRT is associated with significant morbidity, impacting patients' quality of life. Recent advancements in MRI-based risk stratification have raised the possibility of omitting nCRT in selected patients without compromising oncologic outcomes. This study aims to evaluate whether upfront TME alone achieves similar 3-year disease-free survival compared to the standard approach of nCRT followed by TME in patients with cT2N+ and cT3Nx mid-rectal cancer without MRI-based high-risk features regarding local recurrence.

### Methods

The CANO trial is a prospective, multicenter, observational cohort study involving 436 patients across Türkiye. Eligible patients will be classified into two groups: those undergoing direct TME and those receiving nCRT followed by TME. The primary endpoint is 3-year disease-free survival (DFS), with secondary outcomes including 5-year DFS, overall survival, local recurrence rates, and quality of life assessments using validated questionnaires. Data will be prospectively collected and monitored by the steering committee with predefined interim analyses.

### Discussion

The CANO trial addresses the ongoing debate regarding selective omission of nCRT in low-risk mid-rectal cancer. By leveraging MRI-based risk stratification and

**Data availability statement:** No datasets were generated or analysed during the current study. All relevant data from this study will be made available upon study completion.

**Funding:** The Turkish Society of Colon and Rectal Surgery (TSCRS) is the sponsor of the study and covers all the expenses. These include the salary of the secretary employed for data collection, the costs of the server where the database is stored, and the expenses related to the website. TSCRS has also stated that it will cover any future expenses that may arise. The funder has no role in study design, data collection and analysis, decision to publish, or preparation of the manuscript. This statement was included in the manuscript.

**Competing interests:** The authors do not have any financial relationship with the sponsor and receive any personal funding. The authors declare that they have no competing interests. The authors of this protocol and other researchers planned to be involved in the study in the future are members of TRCRS which is the official professional society of colorectal surgeons in Turkey and is a non-profit organization.

a collaborative national network, the study aims to provide high-quality evidence supporting a more personalized treatment approach. The findings have the potential to reduce treatment-related morbidity without compromising oncologic safety, contributing to the refinement of current guidelines.

## Trial registration

ClinicalTrials.gov NCT06823297 [registered before starting inclusion; Version 8, 23.06.2025].

## Introduction

Neoadjuvant chemoradiotherapy (nCRT) followed by total mesorectal excision (TME) has been the standard treatment for locally advanced rectal cancer, significantly reducing local recurrence rates [1–4], particularly in cases with threatened mesorectal fascia (MRF) or a positive circumferential resection margin (CRM) [5,6]. However, despite its oncologic benefits, nCRT is associated with considerable morbidity, including bowel dysfunction, fecal incontinence, and sexual dysfunction, which can severely impact patients' quality of life (QoL) [2,3,7]. Additionally, with advancements in rectal cancer surgery, the necessity of nCRT for all locally advanced rectal cancer cases has been increasingly debated.

Several studies, including OCUM, MERCURY, and QuickSilver, have investigated more selective approaches to nCRT to minimize overtreatment while maintaining oncologic safety [8–10]. These studies have demonstrated that MRI-predicted negative CRM strongly indicates favorable oncologic outcomes, allowing for primary surgery without nCRT in appropriately selected patients. The OCUM trial, in particular, showed that patients with magnetic resonance imaging (MRI)-negative CRM undergoing upfront TME had an acceptably low local recurrence rate (<10%) at five years [8]. The MERCURY trial further validated MRI as a reliable tool for preoperative risk stratification, demonstrating a high accuracy in predicting negative CRM and a low risk of local recurrence in MRI-negative patients [9]. Despite these promising results, the optimal selection criteria for omitting nCRT remain a subject of ongoing research.

Current European guidelines, including those from the European Society for Medical Oncology (ESMO), advocate a risk-adapted approach to nCRT. According to ESMO, patients with cT3a/b rectal tumors located above the levator muscles, without CRM involvement or extramural vascular invasion, are considered to have a low risk of local recurrence and may be candidates for upfront surgery without nCRT [11]. This contrasts with North American guidelines, which still recommend routine nCRT for stage II and III rectal cancers [12]. However, given that modern surgical techniques have significantly reduced the risk of CRM positivity and local recurrence, a more selective approach to nCRT may be warranted.

This trial aims to investigate whether avoiding nCRT provides oncological outcomes equivalent to the standard approach in patients with locally advanced mid-rectal cancer who have a low risk of local recurrence. By leveraging MRI-based risk

stratification and a real-world, multi-center cohort, this study seeks to refine patient selection criteria for nCRT and reduce unnecessary exposure to chemoradiotherapy in low-risk patients.

## Methods

### Study setting

The CANO protocol has been designed by six principal investigators (steering committee) affiliated with five tertiary referral hospitals in Türkiye (Baskent University Hospital, Istanbul Health and Technology University, Dokuz Eylul University Hospital, Halic University Memorial Sisli Hospital and Acibadem Kent Hospital) which are specialized in the treatment of colorectal cancer. The study is coordinated by the study chair (FK) from Baskent University. The steering committee includes two members responsible for writing and updating the protocol (CA, TB), two members responsible for study secretariat and communication with participant centers (IEB, OB), and two members responsible for data storage, reliability, monitoring and safety (TB, AEC).

All centers in Türkiye implementing a multidisciplinary approach to rectal cancer will be invited to participate in the study by Turkish Society of Colon and Rectal Surgery (TSCRS). The clinical administrative officer or department head of all centers will be reached by e-mail. The details of the protocol have been announced on the webpage prepared by the steering committee (https://arastirma.tkrcd.org.tr/). Centers invited to participate in the study will be informed through this website and will also receive additional information via a webinar organized by TSCRS.

In accordance with the guidelines of the Turkish Ministry of Health – Drug and Medical Devices Agency, centers that agree to participate in the study must submit a letter of support to TSCRS on behalf of the study chair (FK). This support letter must include the permission of the institutional data sharing; the signatures of the supervising researcher and clinical administrative officer/department head. Centers that submit their letters by the specified deadline (1 August 2025 anticipated) will be added to the study sites list as an update in the protocol on clinicaltrials.gov.

### Trial registration and ethical considerations

The study protocol has been prepared in accordance with SPIRIT guidelines (S1 File) and approved by Baskent University Ethical Review Board on 15-06-2025 (Approval number: KA24/461). The trial protocol has been registered before starting inclusion (NCT06823297, ClinicalTrials.gov). The study has not started patient recruitment yet and the anticipated recruitment date is 01.08.2025. The anticipated timeline is as follows: Primary recruitment completion: August, 2030, study completion: August, 2030.

Written informed consent for inclusion in the study will be obtained from all eligible patients prior to any treatment (neoadjuvant treatments or surgery), in conjunction with the treatment consents. A distinct informed consent form for the study, prepared by the steering committee and approved by the ethics committee, exists separately from the institutional informed consent form. The attending surgeon is responsible for providing the patient with comprehensive information regarding the study and securing this consent in written form. Another healthcare provider (nurse, resident, surgeon) will witness the consenting process.

In this study, participant data and biological specimens are not being collected for ancillary studies. However, each institution participating in the study may include statements in their own surgical informed consent forms regarding the use of patient information and specimens in other studies. This responsibility lies with the respective institution and the surgeon.

### Trial design

The CANO trial is a prospective, non-randomized, observational, cohort study. Patients undergoing TME will be included in the study, with a comparative analysis between those receiving neoadjuvant chemoradiotherapy and those who do not. The design precludes any possibility of cross-over between the groups.

## Main question

$H_0$: There is no difference in 3-year disease-free survival between direct TME and TME following nCRT in patients with mid-rectal cancer (cT2N+ and cT3Nx) without high-risk MRI findings regarding local recurrence.

$H_1$: Direct TME is associated with worse 3-year disease-free survival compared to TME following nCRT In patients with mid-rectal cancer (cT2N+ and cT3Nx) without without high-risk MRI findings regarding local recurrence.

## Eligibility criteria

The study population consists of male and female patients aged 18 years and older who will undergo TME for histologically confirmed rectal cancer staged as locally advanced (cT2N0-T3N0-N+), without distant metastases or high-risk features including as MRF involvement, EMVI, pathologic lateral pelvic lymph nodes (LPLN) or invasion of adjacent organs (cT4).

### Inclusion criteria.

- Pathologically confirmed rectal cancer
- Rectal cancer within 6–12 cm from anal verge confirmed by sigmoidoscopy or located between the anorectal junction and peritoneal reflection identified by MRI
- Clinically staged with pelvic MRI and thoracoabdominal computed tomography (CT)
- Local clinical stages cT2N+, cT3N0 and cT3N+
- Patients without MRF involvement assessed by MRI (≤1 mm)
- Patients without pathological LPLN (short axis ≥7 mm) on MRI
- Patients without EMVI on MRI
- Patients who underwent open, laparoscopic or robotic TME

### Exclusion criteria.

- cT4 tumors assessed by MRI
- Stage IV disease assessed by MRI and/or CT
- Patients with MSI (+) in TME pathology
- Patients who received neoadjuvant immunotherapy
- Emergency surgery
- Clinical obstruction
- Previous pelvic radiotherapy
- Patients treated without a multidisciplinary council decision
- Inflammatory bowel diseases (Crohn's disease, Ulcerative colitis)
- Familial adenomatous polyposis (FAP), attenuated FAP, and other polyposis syndromes
- Hereditary non-polyposis colorectal cancer (Lynch syndrome)
- Synchronous colon tumors

Patients undergoing surgeries other than TME (e.g., partial mesorectal excision, rectosigmoid resection, end colostomy) will not be included in the study. Following data entry, patients for whom pathology reports indicate that TME was not performed or who has missing data regarding inclusion criteria will be excluded from the study. Patients who are unable to complete neoadjuvant therapy for any reason (such as toxicity, non-compliance, or accessibility issues) will not be excluded from the study. These patients will be evaluated within the neoadjuvant therapy group. Accordingly, when analyzing this aspect, an intention-to-treat analysis will be conducted without cross-over.

### Interventions

The standard of care for the treatment of locally advanced rectal cancer is TME. In patients with cT2N+, cT3N0, and cT3N+ mid-rectal tumors, the administration of nCRT, as well as the direct TME, are both considered guideline-concordant standard treatment approaches and are widely practiced [11,12]. In this study, data from patients undergoing TME as part of routine medical care will be collected prospectively. No additional interventions, treatments and follow-up protocols outside of standard care will be implemented. At the end of the study, patients will be categorized into two groups based on whether they received nCRT and the two groups will be compared in terms of primary and secondary endpoints. No treatment or monitoring protocols differing from routine practice will be proposed for patients or surgeons in the study. The natural prognosis of the treatment arms will be observed.

All the patients undergoing TME described as a precise dissection of the rectum along with the surrounding mesorectal tissue and all pararectal lymph nodes are eligible for the study [13].

**Direct TME Group:** Patients who undergo direct TME without neoadjuvant treatment.

**nCRT Group:** Patients who receive nCRT prior to TME.

Neoadjuvan CRT treatments include all conventional chemoradiotherapy regimens or total neoadjuvant chemoradiotherapy regimens (standard short or long course radiation and/or any standard chemotherapy regimen) described in European and American guidelines [11,12]. The details of radiotherapy and chemotherapy treatments will be recorded, including dosage, frequency, duration, complications, completion status, and the waiting period between treatments. Routine follow-up information for all patients including morbidity, radiologic assessments, recurrence, mortality be recorded in the database for the first five years post-surgery. Patient-reported quality of life scales will be recorded at baseline (prior to the initiation of treatment) and subsequently at 1, 3, and 5 years.

**Restaging.** Detailed data regarding the restaging process for patients receiving neoadjuvant treatment will be systematically recorded in the database. Specifically, the start and end dates of neoadjuvant treatments, as well as the dates of restaging MRI and CT scans, will be documented for each patient, in line with current clinical guidelines. The MRI will evaluate tumor regression, involvement of the mesorectal fascia, and nodal status, while the CT will assess for distant metastases. In this regard, variations between centers can also be evaluated as variables in the regression models during the final analysis.

**Follow-up.** Although follow-up protocols may vary among participating centers, a standardized follow-up scheme will be implemented. All patients will be monitored at regular intervals postoperatively: every three months during the first two years, every six months from years three to five, and annually thereafter. Follow-up assessments will include physical examination, serum carcinoembryonic antigen (CEA) measurement, and imaging studies (chest and abdominal CT, pelvic MRI) as per standard rectal cancer surveillance guidelines. Patient-reported quality of life will be evaluated at 1, 3, and 5 years post-surgery. All follow-up data, including recurrence, survival, and functional outcomes, will be systematically collected and entered into the study database.

### Outcomes

**Primary outcome measure.** 3-year DFS: The proportion of patients who remain free of disease recurrence (local or distant) three years after surgical intervention. DFS will be assessed through clinical evaluations, imaging studies, and pathology reports at regular follow-up intervals.

**Secondary outcome measures.**

- 5-year DFS: The proportion of patients who remain free of disease recurrence (local or distant) five years after surgical intervention.

- 3- and 5-year Overall Survival: The proportion of patients alive at 3- and 5-years post-treatment, regardless of disease status.

- 3- and 5-year Local Recurrence Rate: The percentage of patients experiencing tumor recurrence at the primary site (anastomosis or pelvis) within 3 and 5 years.

- 1-, 3- and 5- year Colorectal Cancer Specific Quality of Life: Patient-reported outcomes assessed using the New Cleveland Clinic Colorectal Cancer Quality of Life Questionnaire [14].

- 1-, 3- and 5- year Bowel Dysfunction Related Quality of Life: Patient-reported outcomes assessed using the low-anterior resection syndrome (LARS) score [15].

Fig 1 represents the timeline of anticipated recruitment and outcome measures.

## Sample size and recruitment

The literature reports a mean 3-year disease-free survival rate of 75–85% for the sample to be included in the study [16]. For patients with mid-rectal tumors (cT2N+ and cT3N0) without high risk MRI findings, the 3-year disease-free survival rate was estimated to be 80% for both direct TME and TME following neoadjuvant chemoradiotherapy. Under these conditions, with α = 0.05 (95% confidence interval) and 80% power, the power analysis indicated that a total of 396 patients (1:1 ratio) would be required. Considering a 10% dropout rate, it is planned to include 218 patients in each group, for a total of 436 patients.

According to data from the Turkish Ministry of Health, General Directorate of Public Health, a total of 24,312 patients were diagnosed with rectal cancer between 2020 and 2024, of whom 17,896 underwent surgery (https://hsgm.saglik.gov.tr/tr/kanser.html). Since this study will be conducted nationally under the sponsorship of the main society of Turkish colorectal surgeons TSCRS, we anticipate reaching the targeted patient number within five years from the start of the study. To achieve this goal, TSCRS will utilize various channels, including mailing, social media, and its website, to announce the study and inform surgeons about the protocol.

## Data collection and management

Researchers will input patient data into the electronic database which is currently used in the TSCRS prospective registry (Redcap). This database includes all variables regarding demographic, clinical, surgical, oncologic and functional outcomes of the patients (S2 File). The patients' baseline clinical tumor stages, clinical tumor stages before and after neoadjuvant treatments, type of surgery, early and late postoperative complications, postoperative pathological tumor stages, imaging results from the routine rectal cancer follow-up protocol, recurrence, and survival data will be prospectively entered into the database by the researchers. Patient reported outcomes will be collected by electronic surveys via TSCRS website. The Turkish versions of New Cleveland Clinic Colorectal Cancer Quality of Life Questionnaire [14] and LARS score [15] have been validated before.

Centers that submit support letters by the specified deadline will have a RedCap username and password created for each researcher in the TSCRS prospective database. Each center can participate in the study with a team consisting of one supervisor/faculty member/specialist doctor and a two-person data collection team (a physician, a medical specialty student, or a medical school student), making a total of three people. The team members are responsible for collecting the data, entering it into the database under the conditions set by the steering committee, and communicating with

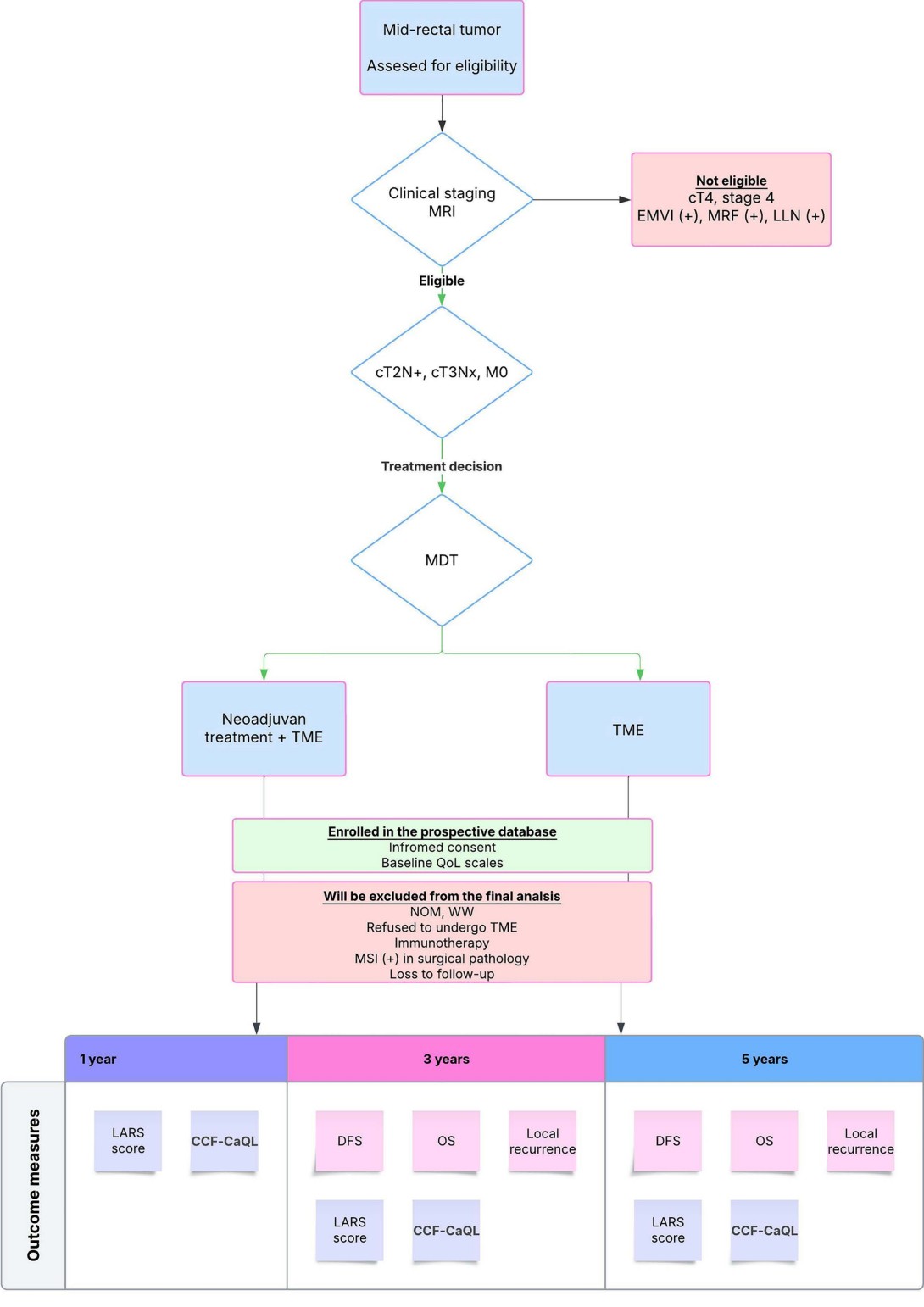

**Fig 1. The timeline showing patient recruitment and measurement timeframes for outcome measures.**

the research secretary when necessary. The supervisor of the team is responsible for the security and accuracy of the data. The entered data will be reviewed and approved by the controller researchers (TB, AEC) and the study secretary (employed by TSCRS) for reliability and errors.

**Access to data and confidentiality.** All protocols have been shared on open-access platforms including clinicaltrials. gov and the website of the trial (https://arastirma.tkrcd.org.tr/). Access to the database containing patient information will be restricted to the steering committee only. The database may be shared with authorities upon request from ethical committees and health authorities. Upon request, de-identified patient information may be shared with third parties and external researchers.

The data entered into the database by the investigators transferred to the SPSS worksheet by the study secretary and stored in a digital cloud system (Microsoft OneDrive) accessible only to steering committee and the study secretary. Consent forms will be delivered to the study secretary by the center supervisors and stored in the cloud by the study secretary. All physical and digital data collected during the study will be retained for 3 years after the completion of the study (a total of 8 years from the start of the study) and will then be permanently destroyed.

## Statistical methods

During the study, missing data analyses and data quality assessments will be actively conducted, and measures to prevent data loss and enhance data quality will be implemented as appropriate. All statistical analyses will be conducted by experts from the TSCRS Biostatistics and Bioinformatics Analysis Unit using SPSS (SPSS Inc, Chicago, IL) software packages.

Multivariable regression analyses will be conducted to evaluate the effect of treatment strategy (direct TME vs. nCRT followed by TME) on the primary and secondary outcomes. The following covariates will be included in the models based on clinical relevance and literature review: age, sex, clinical T stage (cT2 vs. cT3), clinical N stage (N0 vs. N+), tumor differentiation, regimens of neoadjuvant treatment (long-course CRT, short-course RT, TNT with long or short-course RT), interval between neoadjuvant RT/CT and surgery, comorbidities (e.g., Charlson Comorbidity Index), performance status, postoperative complications (Clavien-Dindo), tumor histology, pathologic TME quality (pathologic completeness of the mesorectum), and center effect (as a random effect if appropriate).

For time-to-event outcomes such as disease-free survival (DFS) and overall survival (OS), Cox proportional hazards regression models will be used to estimate hazard ratios and 95% confidence intervals. The proportional hazards assumption will be tested using Schoenfeld residuals.

To address potential selection bias due to the non-randomized design, propensity score methods will be applied. Specifically, propensity scores for receipt of nCRT (vs. direct TME) will be estimated using logistic regression including the aforementioned covariates. The primary analysis will adjust for the propensity score as a covariate in the Cox model. Sensitivity analyses will be conducted using propensity score matching (1:1 nearest neighbor matching without replacement) and inverse probability of treatment weighting (IPTW) to ensure the robustness of results.

Patients lacking the main data specified in the inclusion and exclusion criteria, demographic data, or follow-up data related to the primary endpoint will be completely excluded from the study. Patients with missing data from other variables or secondary endpoints will only be excluded from those specific analyses, and secondary endpoints will be evaluated using data from patients with complete datasets.

## The data monitoring committee and interim analyses

Two members of the steering committee (TB, AEC) are responsible for data monitoring. These investigators will coordinate with the data processing secretary employed by the sponsor (TSCRS) to check the reliability and completeness of the data on a weekly basis. They will present reports to the steering committee every month and to the participant investigators twice a year. They are responsible for interim analysis and adverse event monitoring every six months.

The interim results of the study will be shared at national and international meetings and conferences as the primary and secondary endpoints are reached (in 1, 3 and 5 years after intervention). After completion of the study, the results will be prepared as a manuscript and published in a journal as well as clinicaltirals.gov.

The data monitoring committee members are affiliated with the sponsor, TSCRS, which is a non-profit professional association, and all investigators who are going to be invited in this study are members of this association. An independent monitoring committee has not been established for the study.

**Adverse event reporting, safety and criteria for discontinuing.** In this study, a non-standard treatment that is not currently implemented will not be evaluated. Consequently, side effects and adverse events will be documented as part of the routine monitoring process. However, the steering committee (TB, AEC) will assess interim analyses regarding any safety measures every six months. The study will be terminated if a new treatment relevant to patient groups in rectal cancer management is introduced during the study period and included in the guidelines, and/or there are unacceptable disease-free survival or local recurrence rates (a difference of more than 10%) in any treatment arm that could adversely affect the success of the project. To ensure the safe conduct of the project in the event of such risks, measures will be taken and monitored by the steering committee. The TSCRS will inform relevant parties and all members through e-mail, society website and social media accounts regarding important protocol amendments.

The study will continue until the necessary number of patients is reached. The study will be terminated if a new treatment relevant to patient groups in rectal cancer management is introduced during the study period and included in the guidelines, and/or there are unacceptable disease-free survival or local recurrence rates (a difference of more than 10%) in any treatment arm.

## Discussion

In line with current recommendations, some surgeons perform upfront TME for patients with T2-3 node-positive mid-rectal cancer in the absence of MRF involvement. However, in these cases, the common approach is to administer neoadjuvant chemoradiotherapy. This study seeks to evaluate whether upfront TME achieves non-inferior 3-year disease-free survival compared to the standard.

### Limitations

The non-randomized, observational design introduces the potential for selection bias. Although propensity score methods will be used to minimize selection bias, residual confounding may persist. Our study is expected to span at least five years, which presents logistical challenges both in terms of patient recruitment and follow-up. By the end of this period, the relevance and generalizability of the results may be limited due to potential changes in clinical practice and evolving treatment standards.

Moreover, the trial's reliance on MRI-based risk stratification may lead to variability in interpretations across different centers, particularly in the absence of centralized imaging review. Another limitation is the absence of central pathology review, which may impact the consistency of histopathological assessments. To minimize these limitations, standard MRI and pathology assessments have been published on the study website. Additionally, standardized MRI and pathology evaluation protocols were presented during a webinar organized for candidate participating centers. Furthermore, the TSCRS commits to providing central review and support in cases where centers require assistance with MDT, MRI, or pathology evaluations.

Unfortunately, MSI testing is not yet performed as a standard procedure in endoscopic pathology in Türkiye. Many centers offer it only as an additional test, which is time-consuming, often not covered by insurance, and results in extra costs for patients. Therefore, MSI is tested in endoscopic pathology only if it is routinely performed at the center or if the patient consents to undergo testing at another institution at an additional cost. For this reason, we could not specify unknown pre-treatment MSI status as an exclusion criterion. However, MSI testing is routinely performed in final surgical pathology

in our country. Patients found to have MSI positivity in surgical pathology will not be included in the final analysis. Similarly, patients who underwent preoperative MSI testing (if performed), were found positive, and/or received immunotherapy will also be excluded from the final analysis.

### Strengths

The CANO trial presents several notable strengths that contribute to its potential impact on clinical practice and the broader colorectal cancer literature. First, the study's prospective, multicenter design involving numerous institutions across Türkiye ensures a high level of external validity and generalizability. By enrolling a large cohort of patients, the trial will provide robust statistical power. Additionally, the study's observational nature reflects real-world practice, which enhances the applicability of the findings to routine clinical settings.

Another key strength is the rigorous inclusion and exclusion criteria based on MRI-based risk stratification, which aligns with contemporary guidelines advocating for personalized treatment strategies. The comprehensive data collection process, including both oncologic outcomes and patient-reported quality of life measures such as the LARS score and New Cleveland Clinic Colorectal Cancer Quality of Life Questionnaire, allows for a holistic evaluation of treatment efficacy and its impact on patients' well-being.

The study's voluntary, society-led approach fosters national collaboration among colorectal surgeons, potentially establishing a valuable model for future multicenter research initiatives in Türkiye. The transparent protocol submission and early feedback collection from the scientific community further strengthen the study's methodological rigor.

### Conclusion

The CANO trial has the potential to provide pivotal evidence on the selective use of nCRT in mid-rectal cancer. If successful, the results could guide a more personalized approach to rectal cancer treatment, minimizing overtreatment without compromising oncologic outcomes. The study's findings will not only contribute to the literature but also serve as a model for future collaborative research efforts in Türkiye and beyond.

### Supporting information

**S1 File. SPIRIT guideline cheklist.**
(PDF)

**S2 File. Database draft.**
(XLSX)

### Author contributions

**Conceptualization:** Feza Karakayali, Cigdem Arslan, Aras Emre Canda.

**Data curation:** Feza Karakayali, Tayfun Bisgin, Ilknur Erenler Bayraktar, Onur Bayraktar.

**Formal analysis:** Cigdem Arslan, Ilknur Erenler Bayraktar, Onur Bayraktar, Aras Emre Canda.

**Funding acquisition:** Feza Karakayali.

**Methodology:** Cigdem Arslan.

**Project administration:** Cigdem Arslan.

**Resources:** Tayfun Bisgin, Ilknur Erenler Bayraktar, Onur Bayraktar.

**Software:** Tayfun Bisgin.

**Supervision:** Aras Emre Canda.

**Writing – original draft:** Cigdem Arslan.

**Writing – review & editing:** Ilknur Erenler Bayraktar, Onur Bayraktar, Aras Emre Canda.

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
