## [Decision Letter · Decision Letter 0]

13 Jun 2025

Dear Dr. Arslan,

**One of the reviewers has recommended rejection of the manuscript, while the other has suggested minor revisions. After considering both sets of comments, we believe that the paper has the potential to be reconsidered for publication, provided that the concerns raised by both reviewers are thoroughly addressed.**

**We therefore invite you to revise and resubmit your manuscript, incorporating the suggestions and addressing the issues raised by both reviewers. Please include a detailed response to each comment in your resubmission.**

We look forward to receiving your revised manuscript.

Kind regards,

Alessandra Castelluccia, M.D.

Academic Editor

PLOS ONE

**Journal Requirements:**

1. When submitting your revision, we need you to address these additional requirements. Please ensure that your manuscript meets PLOS ONE's style requirements, including those for file naming. The PLOS ONE style templates can be found at https://journals.plos.org/plosone/s/file?id=wjVg/PLOSOne_formatting_sample_main_body.pdf and https://journals.plos.org/plosone/s/file?id=ba62/PLOSOne_formatting_sample_title_authors_affiliations.pdf 2. Thank you for stating the following financial disclosure: The Turkish Society of Colon and Rectal Surgery (TSCRS) is the sponsor of the study and covers all the expenses. These include the salary of the secretary employed for data collection, the costs of the server where the database is stored, and the expenses related to the website. TSCRS has also stated that it will cover any future expenses that may arise.   Please state what role the funders took in the study.  If the funders had no role, please state: "The funders had no role in study design, data collection and analysis, decision to publish, or preparation of the manuscript." If this statement is not correct you must amend it as needed. Please include this amended Role of Funder statement in your cover letter; we will change the online submission form on your behalf. 3. Please remove all personal information, ensure that the data shared are in accordance with participant consent, and re-upload a fully anonymized data set.  Note: spreadsheet columns with personal information must be removed and not hidden as all hidden columns will appear in the published file. Additional guidance on preparing raw data for publication can be found in our Data Policy (https://journals.plos.org/plosone/s/data-availability#loc-human-research-participant-data-and-other-sensitive-data) and in the following article: http://www.bmj.com/content/340/bmj.c181.long. 4. We note that there is identifying data in the Supporting Information file “S6 database draft”. Due to the inclusion of these potentially identifying data, we have removed this file from your file inventory. Prior to sharing human research participant data, authors should consult with an ethics committee to ensure data are shared in accordance with participant consent and all applicable local laws. Data sharing should never compromise participant privacy. It is therefore not appropriate to publicly share personally identifiable data on human research participants. The following are examples of data that should not be shared: -Name, initials, physical address-Ages more specific than whole numbers-Internet protocol (IP) address-Specific dates (birth dates, death dates, examination dates, etc.)-Contact information such as phone number or email address-Location data-ID numbers that seem specific (long numbers, include initials, titled “Hospital ID”) rather than random (small numbers in numerical order) Data that are not directly identifying may also be inappropriate to share, as in combination they can become identifying. For example, data collected from a small group of participants, vulnerable populations, or private groups should not be shared if they involve indirect identifiers (such as sex, ethnicity, location, etc.) that may risk the identification of study participants. Additional guidance on preparing raw data for publication can be found in our Data Policy (https://journals.plos.org/plosone/s/data-availability#loc-human-research-participant-data-and-other-sensitive-data) and in the following article: http://www.bmj.com/content/340/bmj.c181.long. Please remove or anonymize all personal information (<specific identifying information in file to be removed>), ensure that the data shared are in accordance with participant consent, and re-upload a fully anonymized data set. Please note that spreadsheet columns with personal information must be removed and not hidden as all hidden columns will appear in the published file.

Reviewers' comments:

Reviewer's Responses to Questions

**Comments to the Author**

1. Does the manuscript provide a valid rationale for the proposed study, with clearly identified and justified research questions?

Reviewer #1: No

Reviewer #2: Yes

2. Is the protocol technically sound and planned in a manner that will lead to a meaningful outcome and allow testing the stated hypotheses?

Reviewer #1: No

Reviewer #2: Yes

3. Is the methodology feasible and described in sufficient detail to allow the work to be replicable?

Reviewer #1: No

Reviewer #2: Yes

4. Have the authors described where all data underlying the findings will be made available when the study is complete?

Reviewer #1: No

Reviewer #2: Yes

5. Is the manuscript presented in an intelligible fashion and written in standard English?

Reviewer #1: Yes

Reviewer #2: Yes

You may also provide optional suggestions and comments to authors that they might find helpful in planning their study.

**Reviewer #1: ** Dear authors,

The aim of minimising toxicity in rectal cancer treatment by avoiding unnecessary radiotherapy or chemotherapy is a good one.

Unfortunately, your non-randomised approach does not seem accurate enough to provide robust data.

The non-randomised decisions made by the 'multidisciplinary council' will lead to unadjustable bias.

Furthermore, your inclusion criteria did not recognise risc parameters such as EMVI or lateral lymph nodes.

You do not perform MSS/MSI testing before treatment what should be mandatory.

As surgery seems to be mandatory in your trial, the trend towards organ preservation is not acknowledged in your trial.

You did not describe whether restaging will be performed after NCRT.

In addition, the follow-up scheme, even if it is considered standard, should be described in your paper.

**Reviewer #2: ** Thank you for the opportunity to review the study protocol entitled “CANO trial.” This prospective, multicenter, observational study aims to compare outcomes between upfront total mesorectal excision (TME) and TME following neoadjuvant treatment in patients with cT2N+ or cT3Nany mid-rectal cancer. The protocol is well-structured and addresses a clinically relevant question.

While the overall design is commendable, I have a few methodological concerns that should be clarified:

Minor Comments:

1. MRI Evaluation Consistency:

Given the central role of MRI in risk stratification, the protocol should address how inter-institutional variability in MRI interpretation will be minimized. Consider including provisions such as standardized educational sessions or the use of reference image sets to ensure consistency across sites.

2. Statistical Methods:

Further details are warranted regarding the planned multivariable regression analysis. Specifically, please clarify:

• The covariates to be included in the model

• Whether a Cox proportional hazards model will be used

• How the two treatment strategies will be compared—e.g., via propensity score adjustment, matching, or inverse probability weighting

3. Heterogeneity in Neoadjuvant Treatment (nCRT vs TNT):

The inclusion of both conventional neoadjuvant chemoradiotherapy (nCRT) and total neoadjuvant therapy (TNT) could introduce heterogeneity in the analysis. As TNT has been associated with improved DFS in some studies, how do the authors plan to account for or stratify the impact of TNT separately from standard nCRT?

**Do you want your identity to be public for this peer review?** For information about this choice, including consent withdrawal, please see our Privacy Policy

Reviewer #1: No

Reviewer #2: No

---

## [Author Response · Author response to Decision Letter 1]

23 Jun 2025

Responses to the Comments

Editor#

Dear Editor,

We have reviewed the feedback from you and the reviewers. Taking all feedback into account, we have made several major revisions to our study, which are outlined below.

1. We changed the study design from a non-inferiority design to an observational cohort study. Upon concluding that collecting data from nearly 2000 patients within five years would not be feasible, we conducted a new power analysis. These changes have been updated in the manuscript and the clinicaltrials.gov registry. The ethics committee re-evaluated and approved this new design.

2. We removed the term “non-inferiority” from the study title. Accordingly, all related documents have been updated, and new approvals have been obtained from the ethics committee. We are including the latest versions of these documents in our submission.

3. Regarding the reviewers’ comments: EMVI and lateral lymph node status were already part of our dataset and were being recorded. We have now included these as inclusion criteria as well.

4. The reviewers expressed concerns about multidisciplinary tumor boards and radiology. Our study website already includes a frequently asked questions (FAQ) section for participants. This section contains examples of standardized MR and pathology reports. It also states that support will be provided by the steering committee members’ centers for multidisciplinary board consultations and second opinions on MR and pathology, if requested by participating centers. We have now explicitly mentioned this in the manuscript as well.

Sincerely,

Cigdem Arslan, MD

Reviewer #1:

Dear reviewer,

Thank you for your substantial contributions. We have attempted to revise our protocol accordingly. Please find detailed explanations below. We hope that we have been able to address the limitations of our study.

Kind regards

Cigdem Arslan, MD

1- The aim of minimizing toxicity in rectal cancer treatment by avoiding unnecessary radiotherapy or chemotherapy is a good one. Unfortunately, your non-randomised approach does not seem accurate enough to provide robust data. The non-randomized decisions made by the 'multidisciplinary council' will lead to unadjustable bias.

Response: We fully acknowledge this limitation and have strengthened our statistical analysis plan accordingly. Statistical methods have been revised in the manuscript and highlighted in yellow.

To address potential selection bias, we will estimate propensity using logistic regression with relevant covariates and adjust for the propensity score in the primary Cox regression analysis. Additionally, we will conduct sensitivity analyses using both propensity score matching (1:1 nearest neighbor) and inverse probability of treatment weighting (IPTW) to further mitigate bias and ensure the robustness of our results. Although this cannot completely eliminate bias, we believe that these methods can minimize this limitation in an observational study aiming to reflect real-world data.

2- Furthermore, your inclusion criteria did not recognize risk parameters such as EMVI or lateral lymph nodes.

Response: We assumed that, in the presence of these factors, the MDT would routinely decide on neoadjuvant therapy. However, you are right, and we have now included these factors in the inclusion and exclusion criteria.

3- You do not perform MSS/MSI testing before treatment what should be mandatory.

Response: Unfortunately, MSI testing is not yet performed as a standard in endoscopic pathology in Türkiye. Many centers offer it as an additional test, which is time consuming and usually not covered by insurance and results in extra costs for patients. Therefore, MSI is only tested in endoscopic pathology if the center routinely tests it or if the patient consents to be tested in another institution with extra costs. For this reason, we could not specify unknown pre-treatment MSI status as an exclusion criterion.

However, MSI testing is routinely performed in final surgical pathology in our country. Patients found to have MSI (+) in surgical pathology will not be included in the final analysis. Similarly, patients who had preoperative MSI testing (if performed), were found positive, and/or received immunotherapy will also be excluded from the final analysis. We have revised our flow chart and protocol accordingly to reflect this approach. Also, we have discussed these issues in limitations section.

4- As surgery seems to be mandatory in your trial, the trend towards organ preservation is not acknowledged in your trial.

5- You did not describe whether restaging will be performed after NCRT. In addition, the follow-up scheme, even if it is considered standard, should be described in your paper.

Response: Our database includes restaging and follow-up data and the dates of the examinations detailly. These issues have been added to the methods section in separate paragraphs in the revised manuscript. Thank you.

Reviewer #2:

Dear reviewer,

Thank you for your valuable comments. We have tried to address all in the revised version

Kind regards

Cigdem Arslan, MD

1. MRI Evaluation Consistency:

Given the central role of MRI in risk stratification, the protocol should address how inter-institutional variability in MRI interpretation will be minimized. Consider including provisions such as standardized educational sessions or the use of reference image sets to ensure consistency across sites.

Response: In our study, we aim to observe real-world outcomes. Due to logistical challenges, we are unable to perform centralized radiological and pathological assessments. However, our study website includes a frequently asked questions (FAQ) section where the standards for pathology and MRI reports are clearly described (https://arastirma.tkrcd.org.tr/cano).

Furthermore, during the national Zoom meeting held on June 23, 2025, with centers interested in participating, the radiological and pathological standards were shared with all potential participants. It was also emphasized during the meeting—and published on the website—that centers unable to meet these standards would be offered independent radiologic/pathologic evaluation and MDT (support by TSCRS. We are including this statement in our protocol as well. Thank you.

2. Statistical Methods:

Further details are warranted regarding the planned multivariable regression analysis. Specifically, please clarify:

• The covariates to be included in the model

• Whether a Cox proportional hazards model will be used

• How the two treatment strategies will be compared—e.g., via propensity score adjustment, matching, or inverse probability weighting

Response: Thank you very much for your valuable and constructive comments. We have rewritten the statistical analysis section in the manuscript. Briefly;

Age, sex, clinical T and N stage, tumor differentiation, neoadjuvant treatment regimen, interval between RT and surgery, comorbidities, performance status, postoperative complications, tumor histology, pathologic TME quality, and center effect. Cox proportional hazards models will be used for time-to-event outcomes (DFS, OS). Direct TME vs. nCRT+TME will be compared using propensity score adjustment in the primary analysis, with sensitivity analyses using propensity score matching and IPTW.

3. Heterogeneity in Neoadjuvant Treatment (nCRT vs TNT):

The inclusion of both conventional neoadjuvant chemoradiotherapy (nCRT) and total neoadjuvant therapy (TNT) could introduce heterogeneity in the analysis. As TNT has been associated with improved DFS in some studies, how do the authors plan to account for or stratify the impact of TNT separately from standard nCRT?

Response: In the final analysis, we will also perform subgroup analyses comparing different neoadjuvant treatment regimens with each other.

---

## [Decision Letter · Decision Letter 1]

21 Oct 2025

Can neoadjuvant chemoradiotherapy be omitted in cT2N+ and cT3 mid-rectal cancer: Protocol for a prospective observational cohort study (CANO)

PONE-D-25-12918R1

Dear Dr. Arslan,

We’re pleased to inform you that your manuscript has been judged scientifically suitable for publication and will be formally accepted for publication once it meets all outstanding technical requirements.

Kind regards,

Alessandra Castelluccia, M.D.

Academic Editor

PLOS ONE

Additional Editor Comments (optional):

Reviewers' comments:

Reviewer's Responses to Questions

**Comments to the Author**

1. Does the manuscript provide a valid rationale for the proposed study, with clearly identified and justified research questions?

Reviewer #2: Yes

Reviewer #3: Yes

2. Is the protocol technically sound and planned in a manner that will lead to a meaningful outcome and allow testing the stated hypotheses?

Reviewer #2: Partly

Reviewer #3: Yes

3. Is the methodology feasible and described in sufficient detail to allow the work to be replicable?

Reviewer #2: Yes

Reviewer #3: Yes

4. Have the authors described where all data underlying the findings will be made available when the study is complete?

Reviewer #2: Yes

Reviewer #3: Yes

5. Is the manuscript presented in an intelligible fashion and written in standard English?

Reviewer #2: Yes

Reviewer #3: Yes

You may also provide optional suggestions and comments to authors that they might find helpful in planning their study.

Reviewer #2: Thank you for the comments and replies to my review.

The authors responded to all the comments appropriately.

Although there are several limitations in this study, it may provide valuable information.

Reviewer #3: authors have provided satisfactory responses to reviewers concerns.

please consider:

-add discussion of possible adjuvant treatments for pathology findings of positive resection margins, positive lymph nodes, EMVI, incomplete TME

**Do you want your identity to be public for this peer review?** For information about this choice, including consent withdrawal, please see our Privacy Policy

Reviewer #2: No

Reviewer #3: No

---

## [Editor Report · Acceptance letter]

PONE-D-25-12918R1

PLOS ONE

Dear Dr. Arslan,

I'm pleased to inform you that your manuscript has been deemed suitable for publication in PLOS ONE. Congratulations! Your manuscript is now being handed over to our production team.

Kind regards,

on behalf of

MD Alessandra Castelluccia

Academic Editor

PLOS ONE